# To Code, or Not To Code?
# Exploring Impact of Code in Pre-training

**Viraat Aryabumi, Yixuan Su, Raymond Ma, Adrien Morisot, Ivan Zhang,**
**Acyr Locatelli, Marzieh Fadaee, Ahmet Üstün, Sara Hooker**
`{viraat,ahmetustun,sarahooker}@cohere.com`

## Abstract

Including code in the pre-training data mixture, even for models not specifically designed for code, has become a common practice in LLMs pre-training. While there has been anecdotal consensus among practitioners that code data plays a vital role in general LLMs' performance, there is only limited work analyzing the precise impact of code on non-code tasks. In this work, we **systematically** investigate the impact of code data on general performance. We ask "*what is the impact of code data used in pre-training on a large variety of downstream tasks beyond code generation*". We conduct extensive ablations and evaluate across a broad range of natural language reasoning tasks, world knowledge tasks, code benchmarks, and LLM-as-a-judge win-rates for models with sizes ranging from 470M to 2.8B parameters. Across settings, we find a consistent results that code is a critical building block for generalization far beyond coding tasks and improvements to code quality have an outsized impact across all tasks. In particular, compared to text-only pre-training, the addition of code results in up to relative increase of 8.2% in natural language (NL) reasoning, 4.2% in world knowledge, 6.6% improvement in generative win-rates, and a 12x boost in code performance respectively. Our work suggests investments in code quality and preserving code during pre-training have positive impacts.

## 1 Introduction

The role of data has taken on critical significance in recent breakthroughs. State-of-the-art models highlight the importance of the pre-training data mixture, diversity of data sources (Brown et al., 2020; Longpre et al., 2023; Singh et al., 2024) combined with compute availability as key drivers on performance (Dubey et al., 2024; Üstün et al., 2024; Team et al., 2023; Aryabumi et al., 2024). A critical question is *what properties of data impart the best general performance?*

Perhaps surprisingly, code is often included in pre-training even if a model is not explicitly intended to generate high-quality code. Code datasets differ significantly in terms of structure and textural characteristics from high-quality web datasets (Wikimedia; Raffel et al., 2019). Despite this, several previous generations of LLMs like PaLM (Chowdhery et al., 2022), Gopher (Rae et al., 2022) and Bloom (Workshop et al., 2023) that were not explicitly intended to support code generation, included code data together with high-quality natural language data in their pre-training mixture.

In current state-of-the-art models, it is an accepted norm to not only include code data but further increase the proportion – for instance, Llama 3 (Dubey et al., 2024) has four times more code data in proportion (17%), of its pre-training mixture than Llama 2 (4.5%) (Touvron et al., 2023). While there has been consensus anecdotally among practitioners that code data plays a vital role in LLMs' performance, there has been only limited work analyzing the precise impact of code on non-code tasks. Prior work shows particular side benefits of the inclusion of code data, such as impact on scaling in limited data regime (Muennighoff et al., 2023a), entity tracking capabilities (Kim et al., 2024), and mathematical reasoning (Razeghi et al.). However, there has been no exhaustive study to date that **systematically** investigates the impact of code data on general performance. In this work, we ask "*what is the impact of code data used in pre-training on a large variety of downstream tasks beyond code generation?*".

We embark on an exhaustive set of large-scale controlled pre-training experiments. This includes a consideration of where in the training process adding code is beneficial, code proportions, the role of scaling, and the quality and properties of code added. While a costly endeavor to perform these ablations in a rigorous way, we find consistent and valuable results that code provides critical improvements to non-code performance. In particular, compared to text-only pre-training, for our best variant, the addition of code results in relative increase of 8.2% in natural language (NL) reasoning , 4.2% in world knowledge, 6.6% improvement in generative win-rates, and a 12x boost in code performance respectively. Further performing cooldown with code, improves NL reasoning by 3.7%, World knowledge by 6.8%, and code by 20%, relative to cooldown without code and leads to a 4.1% additional win-rate increase.

Here, several factors matter including getting the proportion of code correct, improving the quality of code by including synthetic code and code adjacent data such as commits, and leveraging code across multiple stages of training including during cooldown. Our results suggest code is a critical building block for generalization far beyond coding tasks and improvements to code quality have an outsized impact on performance. We conduct an extensive evaluation on a broad range of benchmarks, which cover world knowledge tasks, natural language reasoning, and code generation, as well as LLM-as-a-judge win-rates. Across experiments on models ranging from 470 million to 2.8 billion parameter models, we find the following detailed results:

1. **Code provides critical improvements to non-code performance.** Initialization with code pre-trained models results in improved performance for natural language tasks. In particular, compared to text-only pre-training, for our best variant, the addition of code results in a relative increase of 8.2% in NL reasoning, 4.2% in world knowledge, 6.6% improvement in generative win-rates, and a 12x boost in code performance respectively.

2. **Code quality and properties matter.** Using markup-style programming languages, code-adjacent datasets such as GitHub commits and synthetically generated code improves the performance in pre-training. In particular, training on a higher quality synthetically generated code dataset results in a 9% and 44% increase in natural language reasoning and code performance, respectively, compared to web-based code data in pre-training. Additionally, continual pre-training from a code model that includes synthetic data results in 1.9% and 41% relative increases in natural language reasoning and code performance respectively, compared to initialization from a code model that does not include code data.

3. **Code in cooldown enables further improvement across all tasks.** Including code data in pre-training cooldown, where high-quality datasets are up-weighted, leads to an increase of 3.6% in NL reasoning, 10.1% in world knowledge, and 20% in code performance relative to no cooldown. More significantly, cooldown with code beats the baseline (no cooldown) by 52.3% win-rates, where win-rates are 4.1% higher compared to cooldown without code.

## 2 METHODOLOGY

We describe the details of our Pre-training Data (§ 2.1), Evaluation (§ 2.2), Training and Model details (§ 2.3). Figure 1 shows the high-level experimental framework. Precise details for each experiment and their results are presented in Section 3.

### 2.1 PRE-TRAINING DATA

In this section, we describe the details of our pre-training and cooldown datasets. We aim to evaluate the role of code in pre-training, following current state-of-art practices. Hence, we consider pre-training runs that consist of two phases: **1) continued pretraining** and **2) cooldown**. Continued pre-training refers to training a model that is initialized from a pre-trained model and trained for a fixed token budget. Cooldown (Team et al., 2023; Parmar et al., 2024) involves up-weighting high-quality datasets and annealing the learning rate for a relatively small number of tokens during the final stages of training. This up-weighting of high-quality datasets for a smaller amount of steps at the end of training can significantly boost model quality.

**Text dataset.** We use the SlimPajama pre-training corpus (Soboleva et al., 2023) as our source of natural language text data. SlimPajama is a de-duplicated, quality-filtered, multi-corpora, open-source dataset based on RedPajama-1.2T (Computer, 2023). SlimPajama consists of documents

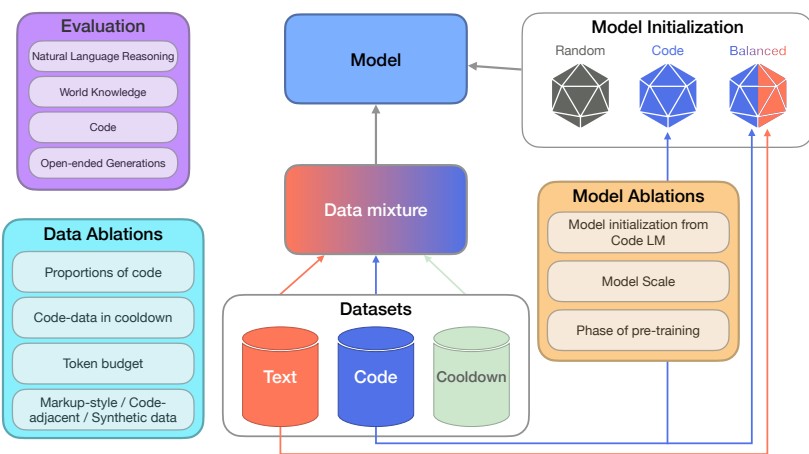

Figure 1: **Overview of our experimental framework**: We exhaustively evaluate the impact of code by varying: 1) the proportion of code in pre-training, 2) code quality and properties, 3) model initialization, 4) model scale, and 5) stage of training at which code is introduced. We evaluate the resulting model on a wide-ranging set of tasks, including natural language reasoning, world knowledge, code, and open-ended generations.

from CommonCrawl, C4, GitHub, Books, ArXiv, Wikipedia, and StackExchange. We filter out all documents from GitHub and StackExchange to remove code and code-adjacent data sources and ensure this is a text-only source. SlimPajama has a total of 627B tokens. After removing all code sources, this results in our text pre-training corpus with a total of 503B tokens.

**Code datasets.** To explore the impact of different properties of code data, we use multiple sources of code in our experiments:

- WEB-BASED CODE DATA: For our main source of code data, we start with the Stack dataset (Kocetkov et al., 2022) that was used to train StarCoder (Li et al., 2023a). The Stack consists of permissively licensed code data scraped from GitHub. We apply quality filters[1] and restrict to the top 25 programming languages based on document count [2]. After all filtering steps, the size of the code-only and markup subset is 139B tokens.

- MARKDOWN DATA We also separately process markup-style languages such as Markdown, CSS, and HTML.[2] After all filtering steps, the size of this markup subset is 180B tokens.

- SYNTHETIC CODE DATA: To ablate the quality of the code dataset, we use a proprietary synthetically generated code dataset that consists of Python programming problems that have been formally verified. We treat this as a high-quality source of code data (See the details in § 3.4). The final synthetic dataset consists of 3.2B code tokens.

- CODE ADJACENT DATA: Finally, to explore different properties of code data, we include a version of the code data which includes auxiliary data such as GitHub commits, jupyter notebooks, StackExchange threads. For GitHub commits, and jupyter notebooks we use the datasets provided as part of the Stack (Kocetkov et al., 2022). We use the version of StackExchange that is part of SlimPajama (Soboleva et al., 2023). In total we have 21.4B tokens of code-adjacent data.

**Pre-training cooldown datasets.** Cooldown involves up-weighting higher quality datasets for the final steps of pre-training and has been found to improve performance on downstream tasks (Parmar et al., 2024; Team et al., 2023), in particular to impart instruction-following capabilities. We choose a cooldown mixture comprising high-quality text, math, code, and instruct-style text datasets.

---

[1]See Appendix C.1 for details about quality filters

[2]Refer to Appendix C.2, C.3 for the full list of programming and markup-style languages included

## 2.2 EVALUATION

Our goal is to systematically understand the impact of code on general performance, which requires a broad evaluation suite that extends to a large variety of downstream tasks beyond code generation. To achieve this, we evaluate models on benchmarks that are reasonable proxies for model ability on **1)** world knowledge, **2)** natural language reasoning, and **3)** code performance. In addition, we report win-rates as evaluated by an LLM-as-a-judge. Table 1 (Appendix A) shows the full evaluation suite and their respective grouping, along with the metric used.

For **World knowledge**, we use benchmarks to measure knowledge memorization, retrieval, and question answering capability given context. We include Natural Questions Open (Kwiatkowski et al., 2019), and TriviaQA (Joshi et al., 2017) as the datasets. **Natural language reasoning** suite consists of 11 benchmarks that involve natural language based reasoning such as Question Answering, natural language inference (NLI), sentence completion, co-reference resolution, and general intelligence. We include the full list of the constituent benchmarks with references in Table 1. Finally, while our main focus is general performance, we also want to measure any changes to code generation performance. For **Code** benchmarks, we focus on the function completion task where we use HumanEval-Python (Chen et al., 2022) and MBPP (Austin et al., 2021).

We evaluate performance at two scales: 470M to 2.8B parameter models. At 470M scale, model capabilities are limited, thus to ensure fair comparisons, we only compare benchmarks for which all models achieve scores above random similar to Muennighoff et al. (2023a); Lozhkov et al. (2024).

**LLM-as-a-judge win-rates.** In addition to task-specific discriminative performance, to allow for a more holistic view across all performance measures, we also evaluate generative performance using LLM-as-a-judge win-rates. This is particularly valuable given recent work that has shown that as performance on open-ended generations improves, there is deterioration in traditional academic tasks (Üstün et al., 2024; Ouyang et al., 2022; Iyer et al., 2022; Muennighoff et al., 2023c). The use of LLMs-as-a-Judge benchmarks (Fu et al., 2023; Liu et al., 2023; Chiang & yi Lee, 2023; Shimabucoro et al., 2024) has gained traction as an alternative to performing human evaluation, which tends to be laborious and expensive (Wang et al., 2023; Boubdir et al., 2023). LLMs as evaluators compare two completions based upon detailed prompts and are reasonable proxies aligned with human preference (Üstün et al., 2024; Dubois et al., 2024).

We use the Dolly-200 English dataset (Üstün et al., 2024; Singh et al., 2024), which consists of 200 hand-picked examples from the Dolly-15K dataset (Conover et al., 2023). These prompts are open-ended and capture general-purpose non-code use cases making them a valuable proxy for how code impacts more fluid and often open-ended tasks. For our win-rate evaluations, we use Command-R+[3] as the LLM judge. Details about the prompt are provided in Appendix D.

## 2.3 TRAINING AND MODEL DETAILS

We use 470M and 2.8B parameters decoder-only auto-regressive Transformer models (Radford et al., 2019) that are trained with a standard language modeling objective. We use parallel attention layers, (Chowdhery et al., 2022; Wang & Komatsuzaki, 2021), SwiGLU activation (Shazeer, 2020), no biases in dense layers, and a byte-pair-encoding tokenizer with a vocabulary size of 256K. All models are pre-trained using AdamW (Loshchilov & Hutter, 2019) with a max sequence length of 8192, batch size of 512 and a cosine LR schedule with a warmup of 1325 steps.

**Infrastructure.** We use TPU v5e chips (Jouppi et al., 2017) for training and evaluation. All models are trained using Jax (Bradbury et al., 2018) framework. We pre-train 64 models in total. This is an enormous endeavour given the scale and computational resources required. Each pre-training run for 200B tokens takes 4736 TPU-chip hours for 470M and 13824 TPU-chip-hours for 2.8B parameter models. Each cooldown run for 40B tokens takes 1024 TPU-chip hours for 470M models.

## 3 RESULTS AND DISCUSSION

In this section, we will report descriptions and results for each experimental variants. We systematically investigate, **(1)** initializing an LLM with code pre-trained models (§ 3.1), and **(2)** the impact of

---

[3]https://huggingface.co/CohereForAI/c4ai-command-r-plus

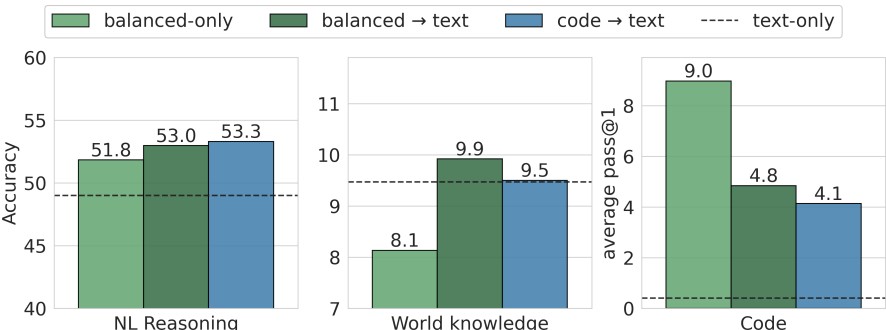

Figure 2: **Impact of initialization using code pre-trained models**: Initializing model training with code pre-trained models improves reasoning and code generation compared to text-only models, where the improvement is the most when continued pre-training with high percentage text (Balanced→Text, Code→Text). Note that these variants are designed to isolate the role of initialization, so do not include cooldown.

model scale (§ 3.2), **(3)** varying proportion of code in pre-training data (§ 3.3), **(4)** quality and properties of the code data (§ 3.4), **(5)** code data in pre-training cooldown (§ 3.5). Finally, we compare the resulting pre-training recipes (§ 3.6). Figure 1 shows the key levers of our experimental design.

## 3.1 INITIALIZING AN LLM WITH CODE PRE-TRAINED MODELS

We explore different initializations of pre-trained models to understand if using an LM with a large portion of code data as initialization improves the performance. These key ablations, along with their token counts, are summarized in Table 2. We briefly describe below:

- **Text LM** (TEXT-ONLY BASELINE): Pre-trained model *from scratch* using glorot-normal initialization (Glorot et al., 2011) on the text-only data for 400B tokens.
- **Balanced LM** (BALANCED-ONLY): A model is trained with an equal ratio of code and text data (50% text and 50% code) in pre-training for 400B tokens.
- **Balance-initialized Text LM** (BALANCED → TEXT): This model is initialized with a balanced LM (50% text and 50% code) and further pre-trained using text data for 200B tokens.
- **Code-initialized Text LM** (CODE → TEXT): Different from other variants, this model is initialized with a code-LM which is pre-trained on a code dataset for 200B tokens. The code dataset contains a mixture of 80% code data and 20% markup-style code data. We then continually pre-train this model on text for another 200B tokens.[4]

**Natural Language Reasoning** As seen in Figure 2, initializing with 100% code pre-trained model (code→text) has the best performance for NL Reasoning benchmarks, and is closely followed by the balanced→text model. The code→text model and balanced→text model beat the text-only baseline on NL reasoning tasks by 8.8% and 8.2% relative improvement respectively. The balanced-only model improves upon the baseline by 3.2%. This shows that initialization from a pre-trained model with a mix of code has a strong positive effect on NL reasoning tasks. Further using a text mix with a small percentage of code for continual pre-training results in the best performance as evidenced by both the code→text and balanced→text models.

**World Knowledge** For World Knowledge tasks, we see that the balanced→text model has the best performance over all other variants, beating the code→text by 21% and text-only by 4.1% relative improvement. This suggests that performance on world knowledge tasks depends on a more balanced data mixture for initialization and a larger proportion of text in the continual pre-training stage. Overall, code data is still beneficial compared to text-only pre-training for world knowledge tasks.

---

[4]We use a 10% of code in text mixture data during continual pre-training of code-initialized models (balanced→text, code→text) to avoid a full distribution shift and maintain the benefits of code.

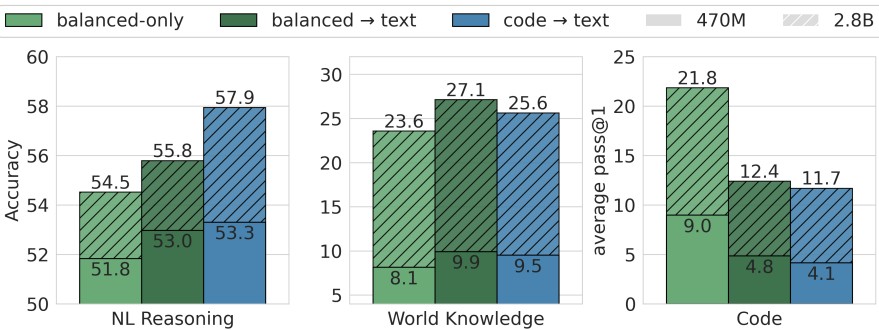

Figure 3: **Impact of model scale on different tasks.** We observe that scale provides pronounced gains across tasks of up-to 2.7x increase, however the overall trend remains the same across scales showing consistency of findings across model sizes.

**Trade-offs between NL tasks and code generation** For code generation, `balanced-only` achieves the best performance, where we see a 46.7% and 54.5% relative improvement over `balanced→text` and `code→text`. This is expected as `balanced-only` includes 50% code throughout pre-training. However, this model trades off better code generation with lower performance in NL tasks. `code→text` and `balanced→text` achieves 2.9% and 2.3% relative increase in NL reasoning, and 17.3% and 22.2% relative increase in world knowledge respectively compared to `balanced-only`.

**Generative quality win-rates comparison** Additionally, we compare the generative performance of each code variant (`code→text` and `balanced-only`) against the `text-only` model. We report win-rates and observe that the presence of code has a strong positive impact on generation quality. Both `code→text` and `balanced-only`) models beat the `text-only` variant by a 6.6% difference in win-loss rates. We again note that Dolly-200-English evaluation set we use for win-rate calculation is curated to reflect open ended questions and is a non-code evaluation. This confirms that code data in the pre-training mix does not *only* improves reasoning but also helps the model produce better quality generations.[5]

## 3.2 IMPACT OF SCALE

To understand if the findings of Section 3.1 transfer to larger models, we train 2.8B parameters models with the same token budget following the same model variants at 470M scale. Figure 3 shows the results of 2.8B models in comparison with 470M results.

**Comparison between 2.8B and 470M models** Scaling model size to 2.8B enables higher performance for all model variants in all task categories, compared to 470M results. In terms of average performance across NL reasoning and world knowledge, `balanced→text` model benefits from scaling-up by a 33.1% increase relative to the same model with 470M size. The improvement for `code→text` and `balanced-only` are 31.7% and 30% relative increase.

We find that the improvements in NL reasoning are relatively modest with 5.3%, 9.2%, and 5.2% relative gains for `balanced→text`, `code→text`, and `balanced-only` respectively. However, world knowledge and code performance nearly triples for all the model variants. In particular, 2.8B `balanced→text` results increase by 2.7x in world knowledge and 2.5x in code evaluation compared to 470M.

**Trends between model variants in 2.8B** Notably, in terms of initialization with code pre-trained models, the same trends seen in 470M parameter scale hold at 2.8B models. `code→text` and `balanced→text` models improve over `balanced` models by 6.9% and 6.1% relative gain, however, fall significantly behind in code generation performance with 43.1% and 46.3% relative drop. These results show that the trade-off between NL tasks and code generation increases with the model size. Overall our experiments scaling to a larger size shows that our results hold and are consistent with the trends we observe at 470M parameter ablations.

---

[5]We include the extended Win-rates for these experiments in Appendix E.

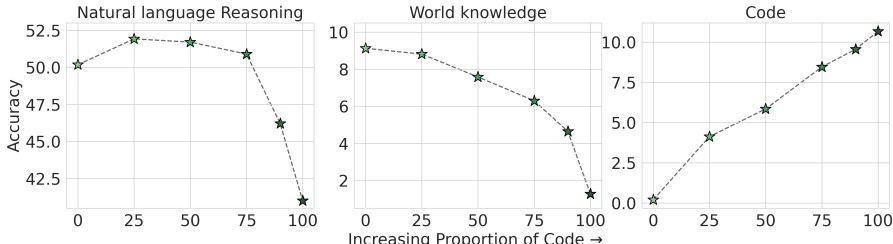

Figure 4: **Impact of the proportion of code in pre-training for different tasks**: We observe that as the code proportion of pre-training increases, the performance on code tasks increases linearly. In contrast, there is more sensitivity for NL reasoning and World knowledge tasks and an optimal range of code proportions where benefits are most tangible. Model size is 470M parameters and trained for 200B tokens.

### 3.3  CODE DATA PROPORTION IN PRE-TRAINING

In these experiments, we ablate the proportions of code data in the pre-training mixture to understand what is the optimal amount of code to maximize performance on non-code tasks. Here, we focus on the first phase of pre-training with random initialization. We train six models for 200B tokens with increasing code proportions: 0%, 25%, 50%, 75%, 90%, and 100%. The remaining proportion is filled with text data. For each variant, we train a new model independently in order to carefully ablate the impact of varying code proportions.

**Natural Language Reasoning and World Knowledge** For NL Reasoning, as the amount of code increases, in Figure 4 we see an increase in performance compared to a text-only (0% code) model. The best performance is from a model with 25% code and 75% text, with a 3.4% relative improvement over a model with 0% code. While performance is maintained up to 75% code, it starts to rapidly erode at higher proportions with a sharp relative drop of 18.3% when the model is trained on 100% code compared to a model with no code.

For World Knowledge tasks, we see an inverse relationship with increasing the amount of code. As seen in Figure 4 middle inset, there is a slight relative drop of 3.4% at 25% code and this relative drop worsens to 31% at 75% code compared to the no-code model. The fully code model (100% code) is unable to perform in world knowledge task (86% drop relative to text-only) as there are no data sources to acquire the required knowledge in the pre-training mix.

**Performance on Code** For code evaluation, there is a linear increase in performance as the amount of code increases, with the best model being a code-only model. As observable in Figure 4 right inset, the 100% code leads to a 2.6x increase in the code benchmarks compared to the 25% code model. As expected, for the model with 0% code, the average pass@1 score drops to 0.

### 3.4  INFLUENCE OF CODE QUALITY AND PROPERTIES ON GENERAL PERFORMANCE

In this section, we investigate the properties of code data by varying its quality and composition. We study this firstly **(a)** from the perspective of training *from scratch*, as we want to isolate the exact effects of different properties of code data. Secondly **(b)**, we incorporate the best variant of the code data (high-quality synthetic code), in our continual pre-training experiments to see if the impact of the code quality transfer. We report performance on NL reasoning and Code tasks.

We study the effect of the following properties: **(1)** MARKUP-STYLE DATA: we separate markup-style programming languages (§ 2.1) from the rest of web-based code (Appendix C.3). We replace 20% of code-only tokens with markup-style tokens. **(2)** CODE ADJACENT DATA: Instead of using purely web-based code data, we replaced 15% percentage of code tokens with code-adjacent datasets - GitHub issues (5%), StackExchange (5%) and Jupyter Notebooks (5%), resulting in a `code-adjacent` model. **(3)** CODE QUALITY: To control the quality of the code, we replaced 10% of existing code tokens with a synthetically generated high-quality code dataset. The remaining proportions of web-based code data are kept the same, resulting in a `code-synth` model.

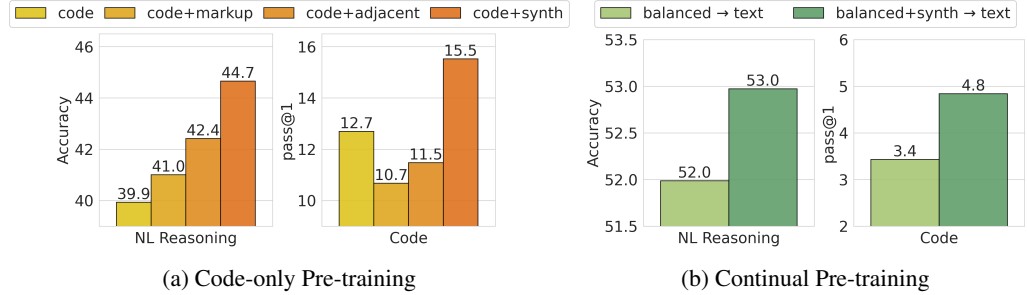

Figure 5: **Impact of using different properties of code data**: **(a)** As the most impactful code data source, synthetically generated high-quality code improves NL reasoning and code performance for code pre-training. **(b)** These improvements with synthetically generated high-quality code data also transfer the continual pre-training setting. All models are of size 470M parameters.

**Code-only pre-training** We compare the above variants to a model that is trained only on web-based code data (`code`) from the stack dataset (Kocetkov et al., 2022), which forms our baseline model. All the variants are pre-trained using the same amount of tokens (200B) for fair comparison.

In Figure 5a, we evaluate the impact of code quality and code composition. We observe that across all variants, including diverse code sources and also synthetic code leads to gains in natural language performance relative to `code`, however, only synthetically generated code improves the code benchmarks. We relate this to our code evaluation where we measure performance in *python*, thus different programming languages or code-adjacent data slightly decrease the results. Here, `code+markup` and `code+adjacent` leads to 2.8% and 6.3% relative improvement in NL reasoning compared to **code** (web-code-only), but cause 15.7% and 9.4% drop in code evaluation.

Our synthetic code data (`code+synth`) is the most impactful ablation. It is particularly impressive given its relatively small share of the overall dataset. Despite a small weighting of 10%, the inclusion of synthetic data leads to relative improvements of 9% on NL reasoning, and 44.9% on code benchmarks compared to the baseline of web-code-only. We note that the lifts observed for synthetic data are even more impressive given the limited amount of synthetic data available compared to code-adjacent data (3.2B tokens vs 21.4B tokens) or code+markup data (3.2B tokens vs 40B tokens), and the weighting during pre-training allocation (10% vs 15% vs 20% for synthetic data, code-adjacent, code-markup respectively). This suggests a key future lever of improvement is increasing the proportion of such high-quality code data sources.

**Continual pre-training** Here, based on the findings from code-only pre-training, we incorporated `code+synth` into our best continual pre-training variant (`balanced+synth→text`). We compare this against the same variant without synthetic code data (`balanced→text`) to evaluate the benefits of synthetic data. We use the same amount of code and text tokens in these experiments.

As shown in Figure 5b, `balanced+synth→text` achieves 2% and 35% relative improvement over `balanced→text` in NL reasoning and code, respectively. This further confirms that even a small percentage of a high-quality code data, not only improves performance in code pre-training but also increases code and non-code performance after continual pre-training with text data.

### 3.5 CODE IN PRE-TRAINING COOLDOWN

In this section, we evaluate the impact of code at the final stage of pre-training. Here, we consider cooldown, where we up-weight high-quality text, math, code, and instruct-style datasets. We change the learning rate schedule from cosine-based to linear annealing with a final learning rate of $1e-6$. We evaluate the impact of code in cooldown by comparing 3 models: a pre-trained model before cooldown, cooldown without code data, and cooldown with 20% code data. For our pre-trained model, we use `balanced→text` as it is our best pre-trained variant. We preserve the same token budget across variants – 40B tokens which is 10% of the token budget of the pre-trained model.

**Impact of code used during cooldown in different tasks** In Figure 6a, we evaluate the impact of code in cooldown on model performance in NL Reasoning, world knowledge, and code benchmarks.

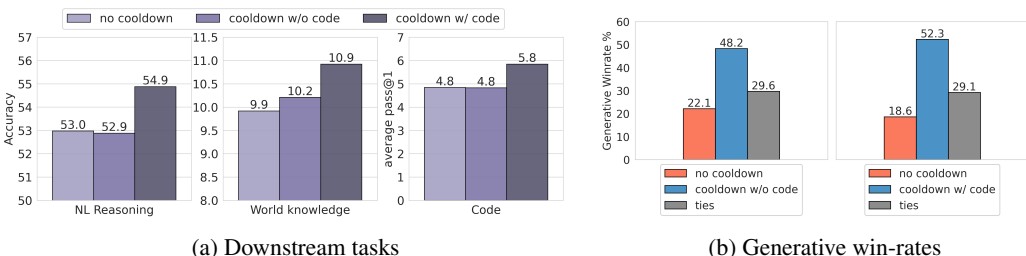

(a) Downstream tasks                    (b) Generative win-rates

Figure 6: **Impact of code data in pre-training cooldown**: Including code data in the cooldown phase improves downstream relative to cooldown with no code. All cooldown variants benefit for downstream tasks and especially generative quality. We find the largest gains from cooldown with code, with the highest win-rates of 52.3 % over a model with no cooldown.

Across tasks, we find that a cooldown with code data is most beneficial with 3.6%, 10.1%, and 20% in NL reasoning, world knowledge, and code relative to the model without cooldown.

In contrast, we find that cooldown without code does not provide any increases for both NL reasoning and Code, while providing a relative improvement of 3.1% in World Knowledge tasks compared to no cooldown, showing the critical role of code data in also cooldown phase of pre-training.

**Generative win-rates after cooldown** As expected, cooldown has a significant impact on generative performance as measured by win-rates (seen in Figure 6b). This is because we up-weight high-quality data sources in pre-training mix including instruction-style datasets such as Dolly v2 (Conover et al., 2023). Both cooldown variants (`cooldown w/o code`, `cooldown w/ code`) beat `no-cooldown` model by large win-rates (48.2% and 52.3%) as seen in Figure 6b. Comparing the cooldown variants, including code leads an additional 4.1% generative win-rates against `no-cooldown` compared to cooldown without code.

## 3.6 COMPARING PRE-TRAINING RECIPES

Considering all our experiments, we summarize our findings and recommend recipes for pre-training with code data. Table 2 (Appendix B) shows the different variants along with pre-training phases.

**Best recipe for natural language tasks** As seen in Sections 3.1, 3.3, and 3.5, including code in all phases of pre-training provides improvements across all task categories. When looking at the final recipes, we find that `balanced→text` model followed by cooldown that includes code data corresponds to the best overall performance in natural language tasks considering NL reasoning, world knowledge, and generative performance. Notably this model achieves the highest generative win-rates with 37.7% vs 33.7 against `text-only` as shown in Figure 7.

**Best recipe for code performance** Among complete recipes shown in Table 2, `balanced-only` provides the best performance in code benchmarks. This model achieves 20% relative gain compared to second best `code→text` and 55% relative gain compared to `balanced→text`. However, `balanced-only` falls behind in natural language performance by 2.5% relative difference and 5.0% win-rate difference (vs `text-only`), both compared to `balanced→text`.

Including code in all phases of pre-training is beneficial across our three task categories and generative performance. Our recommendation for the best overall performance is to include a balanced mixture of code and text data during pre-training from scratch (§ 3.3), use a relatively lower code percentage during continual pre-training (§ 3.1), and include code data into cooldown mixture. Further, we recommend including high-quality code data during all phases of pre-training (§ 3.4).

## 4 RELATED WORK

**Understanding the impact of pre-training mixes** Several works have studied the effects of data age, quality, toxicity and domain of pre-training data (Longpre et al., 2023; Üstün et al., 2024). Several works have looked at the impact of filtering (Raffel et al., 2020; Rae et al., 2021; Penedo et al., 2023), de-duping (Zhang et al., 2022) and data pruning (Lozhkov et al., 2024; Marion et al., 2023;

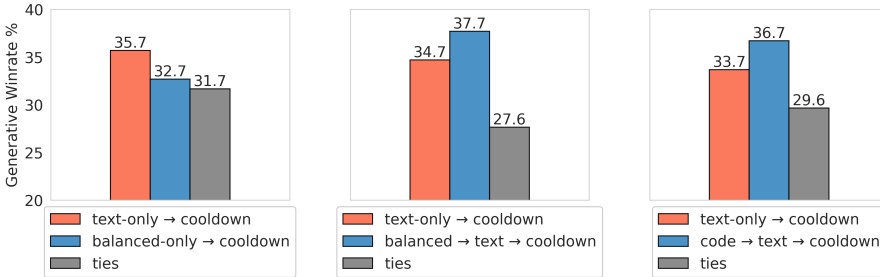

Figure 7: Generative performance as measured by win-rates for variants with full-cooldown.

Chimoto et al., 2024; Boubdir et al., 2023). Furthermore, several works have considered the role of synthetic data at improving performance (Shimabucoro et al., 2024; Dang et al., 2024; Aakanksha et al., 2024) and helping bridge the gap in performance between open weights and proprietary models (Gunasekar et al., 2023; Li et al., 2023b). In contrast to our work which focuses explicitly on understanding the role of code, these studies focus on characteristics of training data as a whole.

**Understanding the role of code** Including code in the pre-training data mixture, even for models not specifically designed for code, has been a common practice in LLMs pre-training (Dubey et al., 2024; Gemini-Team et al., 2024; Groeneveld et al., 2024). In addition to serving the popular use case in code completion and generation (Chen et al., 2021), previous studies suggest that the addition of code improves the performance of LLMs on various NLP tasks, such as entity linking (Kim et al., 2024) and commonsense reasoning (Madaan et al., 2022b)), mathematical reasoning tasks (Liang et al., 2022; Madaan et al., 2022a; Gao et al., 2023; Shao et al., 2024), and general reasoning capabilities (Muennighoff et al., 2023a; Fu & Khot, 2022; Ma et al., 2023). Muennighoff et al. (2023b) demonstrated Python code data can be used to improve pretraining performance. They focused on a low-resource pre-training regime with limited data and an evaluation set-up limited to perplexity evaluations. Zhang et al. (2024) investigated the impact of code on LLMs' internal reasoning capability across various tasks and model families. They only focus on the effect of code in the supervised fine-tuning stage (SFT) primarily measuring the impact on reasoning. Zhu et al. (2024) report the performance of their DeepSeek-Coder-V2 models on General Natural Language benchmarks. They compare chat and instruct models, and do not investigate different phases of pre-training and properties of code.

To the best of our knowledge, this work is the first study that presents a thorough investigation of the impact of code in pre-training on non-code tasks. Our experiment spans several axes and a exhaustive evaluation suite, with costly ablations at scale including model initialization strategies, different proportions and properties of code data, and model scales.

## 5 CONCLUSION

We perform a first-of-its-kind systematic study to answer "*what is the impact of code data used in pre-training on a large variety of downstream tasks beyond code generation*". We focus, not just on code performance but on downstream natural language performance, as well as generative quality using LLM-as-a-judge win-rates. We conduct ablations that look at initialization, proportions of code, quality and properties of code, and role of code in pre-training cooldown. We find across all scales of experiments that code provides critical improvements to performance on non-code tasks. Compared to text-only pre-training, for our best variant, the addition of code results in relative increase of 8.2% in natural language (NL) reasoning, 4.2% in world knowledge, 6.6% improvement in generative win-rates, and a 12x boost in code performance respectively. Further performing cooldown with code, improves 3.6%, 10.1%, and 20% in NL reasoning, world knowledge, and code relative to the model before cooldown and leads 52.3% generative win-rates. Finally, we find that adding a small amount of high-quality synthetic data can have an outsized impact on both NL reasoning (9% relative increase) and code performance (44.9% relative increase).

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

## ETHICS STATEMENT AND LIMITATIONS

While we systematically study the impact of code data on downstream natural language tasks, we do not study its impact on safety and bias. Additionally, given the nature of pre-training and the number of ablations we have conducted we were limited by the scale of larger model sizes due to prohibitive compute costs.

## REPRODUCIBILITY

We provide details about our data mixture (Section 2.1), data filtering (Appendix C.1, C.2, C.3), evaluation (Section 2.2, Appendix A) and training (Section 2.3) setups. We believe these details provide a clear picture on how to obtain our data setup, model ablations and evaluation results.

## A EVALUATION DETAILS

We briefly describe the details of our evaluation benchmarks and the composite datasets used for each category below:

1. **World knowledge.** These benchmarks aim to measure world knowledge, testing knowledge memorization, retrieval, and question answering capability given context. We include Natural Questions Open (Kwiatkowski et al., 2019), and TriviaQA (Joshi et al., 2017) as the datasets. We report the average exact match scores for both these benchmarks.

2. **Natural language reasoning.** The Natural language (NL) reasoning suite consists of 11 benchmarks that involve natural language based reasoning such as Question Answering (Clark et al., 2019; Seo et al., 2018; Welbl et al., 2017; Sap et al., 2019; Choi et al., 2018), natural language inference (NLI) (Wang et al., 2020; de Marneffe et al., 2019; Wang et al., 2020), sentence completion (Mostafazadeh et al., 2016; Zellers et al., 2019), co-reference resolution (Sakaguchi et al., 2019) and general intelligence (Clark et al., 2018). We include a full list of the constituent benchmarks in Table 1. We report the average accuracy scores across all benchmarks.

3. **Code.** While our main focus is general performance, we also want to measure any changes to code generation performance. For code benchmarks, we focus on the function completion task. We evaluate on HumanEval-Python (Chen et al., 2022) and MBPP (Austin et al., 2021). We report the average `pass@1` scores of these benchmarks.

## B SUMMARY RESULTS FOR PRE-TRAINING RECIPES

Summary results are shown in Table 2.

## C CODE-DATASETS FILTERING

### C.1 QUALITY FILTERS

In addition to the deduplication and quality filtering applied on the GitHub scrapes by Starcoder for The Stack dataset (Li et al., 2023a), we apply filters to remove documents with greater than 1000 float numbers, with instances of the string `0x`, that are lists of top-level domains, and with 'generated by' in the first 400 characters

### C.2 PROGRAMMING LANGUAGES PRESENT IN WEB-BASED CODE DATASET

Programming languages included in our version of The Stack dataset are present in Table 3

### C.3 MARKUP-STYLE PROGRAMMING LANGUAGES PRESENT IN WEB-BASED CODE DATASET

Markup-style languages included in our version of The Stack dataset are in Table 4

| Task | Dataset | | Metric |
|------|---------|---|--------|
| **WORLD KNOWLEDGE TASKS** | | | |
| Question Answering | TriviaQA (Joshi et al., 2017) | 0-shot | Acc. |
| | NaturalQuestionsOpen (Lee et al., 2019) | 0-shot | Acc. |
| **NATURAL LANGUAGE REASONING** | | | |
| Question Answering | BoolQ (Clark et al., 2019) | 0-shot | Acc. |
| | PiQA (Seo et al., 2018) | 0-shot | Acc. |
| | SciQ (Welbl et al., 2017) | 0-shot | Acc. |
| | SocialIQA (Sap et al., 2019) | 0-shot | Acc. |
| | QUAC (Choi et al., 2018) | 0-shot | Acc. |
| Natural Language Inference | SuperGLUE-CB (Wang et al., 2020; de Marneffe et al., 2019) | 0-shot | Acc. |
| | SuperGLUE-COPA (Wang et al., 2020) | 0-shot | Acc. |
| Sentence Completion | StoryCloze (Mostafazadeh et al., 2016) | 0-shot | Acc. |
| | HellaSwag (Zellers et al., 2019) | 0-shot | Acc. |
| Coreference Resolution | Winogrande (Sakaguchi et al., 2019) | 0-shot | Acc. |
| General Intelligence | ARC-Easy (Clark et al., 2018) | 0-shot | Acc. |
| **TEXT GENERATION** | | | |
| Open-Ended Generation | Dolly-200 (English) (Singh et al., 2024) | 0-shot | win-rate |
| **CODE GENERATION** | | | |
| Function completion | HumanEval (Chen et al., 2021) | 0-shot | pass@1 |
| | MBPP (Austin et al., 2021) | 0-shot | pass@1 |

Table 1: **Datasets considered for evaluation**: We conduct extensive evaluations across benchmarks detailed above. These provide valuable proxies for performance in natural language reasoning, world knowledge, open ended text generation, and code generation tasks.

| Model Variant | Recipe | Token Count | | Natural Language | | | Code | Total Avg. |
|---------------|--------|------|------|--------|-------|------|------|-----------|
| | | Text | Code | Reason. | Know. | Avg. | | |
| TEXT-ONLY | Pre-training | 400B | - | 49.0 | 9.5 | 29.2 | 0.4 | 19.6 |
| | Cooldown | +32B | +8B | 54.1 | 11.1 | 32.6 | 4.4 | 23.2 |
| BALANCED-ONLY | Pre-training | 200B | 200B | 51.8 | 8.1 | 30.0 | 9.0 | 23.0 |
| | Cooldown | +32B | +8B | 53.2 | 11.1 | 32.1 | 8.4 | 24.2 |
| BALANCED → TEXT | Pre-training Init. | 100B | 100B | 52.0 | 7.4 | 29.6 | 7.8 | 22.4 |
| | Continue Pre-train. | +180B | +20B | 53.0 | 9.9 | 31.5 | 4.8 | 22.6 |
| | Cooldown | +32B | +8B | 54.9 | 10.9 | 32.9 | 5.8 | 23.9 |
| CODE → TEXT | Pre-training Init. | - | 200B | 44.7 | 1.5 | 23.1 | 15.5 | 20.6 |
| | Continue Pre-train. | +180B | +20B | 53.3 | 9.5 | 31.4 | 4.1 | 22.3 |
| | Cooldown | +32B | +8B | 52.1 | 10.3 | 31.2 | 7.5 | 23.3 |

Table 2: **Model variants with the corresponding pre-training recipes**: Pre-training recipes include initial pre-training, continued pre-training, and cooldown phases. Balanced→Text achieves the best NL performance while Balanced-only performs significantly better in code generation.

# D  LLM JUDGE PROMPT AND PREAMBLE FOR WIN-RATES

**Preamble**

```
You are a helpful following assistant whose goal is to select the
preferred
(least wrong) output for a given instruction.
```

**Prompt**

```
Which of the following answers is the best one for the given
instruction.
A good answer should follow these rules:
1) It should have correct reasoning,
2) It should answer the request in the instruction,
3) It should be factually correct and semantically comprehensible,
4) It should be grammatically correct and fluent.

Instruction:  instruction
```

| Language Name | Proportion of total code documents |
|---|---|
| java | 15.54 |
| javascript | 15.29 |
| php | 12.46 |
| python | 9.60 |
| c-sharp | 8.30 |
| typescript | 7.92 |
| c | 6.63 |
| cpp | 4.91 |
| go | 3.49 |
| ruby | 2.69 |
| shell | 1.82 |
| kotlin | 1.76 |
| Swift | 1.52 |
| Vue | 1.48 |
| rust | 1.00 |
| scala | 0.94 |
| JSX | 0.83 |
| sql | 0.74 |
| dart | 0.72 |
| makefile | 0.53 |
| lua | 0.47 |
| haskell | 0.45 |
| smalltalk | 0.43 |
| tex | 0.37 |
| clojure | 0.10 |

Table 3: Programming languages included in our version of The Stack dataset

| Language Name | Proportion of total code documents |
|---|---|
| markdown | 54.23 |
| yaml | 10.77 |
| json | 9.97 |
| html | 8.57 |
| css | 6.86 |
| SCSS | 5.84 |
| restructuredtext | 2.26 |
| TOML | 1.25 |
| rmarkdown | 0.02 |
| Sass | 0.22 |

Table 4: Markup-style languages included in our version of The Stack dataset

```
Answer (A): completion_a

Answer (B): completion_b

FIRST provide a concise comparison of the two answers which
explains
which answer you prefer and why.
SECOND, on a new line, state exactly one of
'Preferred:  Answer (A)' or 'Preferred:  Answer (B)' to indicate
your choice
of preferred response.

Your response should use the format:
Comparison:  <concise comparison and explanation>
Preferred:  <'Answer (A)' or 'Answer (B)'>
```

## E    GENERATIVE WIN-RATES FOR IMPACT OF INITIALIZATION

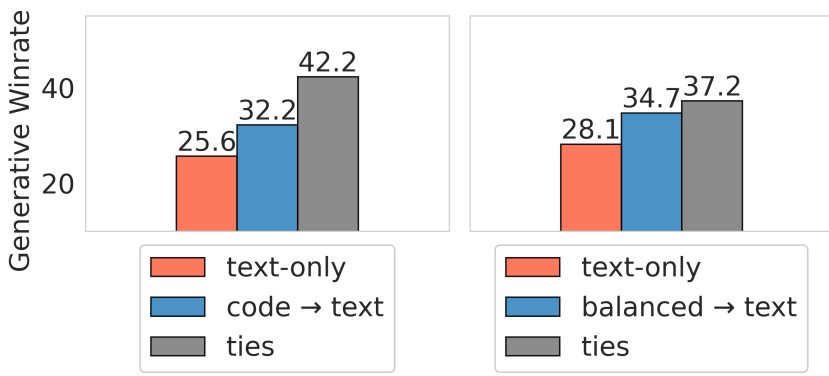

Figure 8: **Impact of initialization on generative quality as judged by LLM-as-a-judge.**

## F    EVALUATION OF 470M COOLDOWN MODELS ON GSM8K

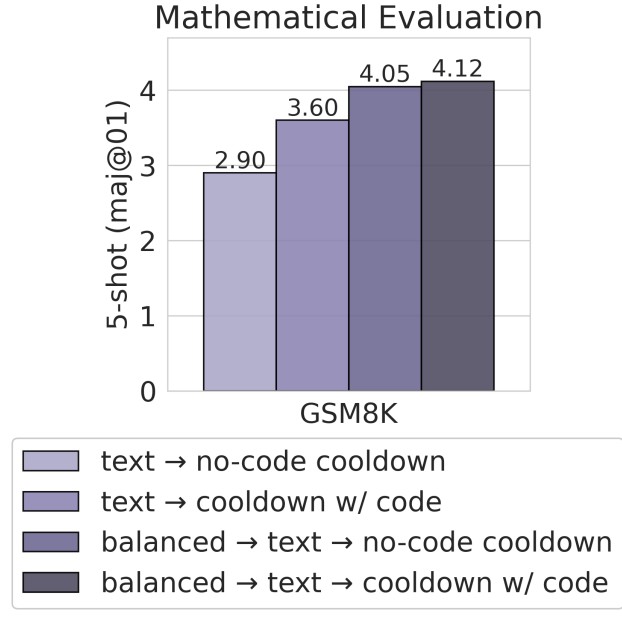

Figure 9: **Evaluation of 470M cooldown models on GSM8K** Including code in any stage of the pre-training improves performance compared to the model where no code has been seen in any of the training stages: pre-training, continual pre-training and cooldown. The most performant model in this comparison has seen code in all stages including cooldown where it leads a significant improvement (from 2.9 to 4.12, +42% relative gain).

