# OpenReview forum: "To Code or Not To Code? Exploring Impact of Code in Pre-training"
_ICLR.cc/2025/Conference — ICLR 2025 Poster_

### Official Review · Reviewer_YwMY · 2024-10-31

**Soundness:** 3
**Presentation:** 3
**Contribution:** 3
**Rating:** 8
**Confidence:** 4

**Summary:**

## Summary:
The authors conduct a systematic investigation into the role of code data in the pre-training phase of LLMs, assessing the impact on three task categories: natural language reasoning, world knowledge, and code generation. They analyze three pre-training stages, including initial pre-training, continual pre-training, and a cooldown phase; evaluate the effect of code data’s proportion, quality, and properties on the model’s performance across the three task categories. Their results demonstrate that incorporating code data enhances model performance across all the pre-training stages, suggesting a balanced approach with optimized code data quality can boost the model’s ability in reasoning, general knowledge, and coding.

**Strengths:**

- _Novelty_: This work is a first-of-its-kind, comprehensive study on how code data in pre-training affects LLMs across various natural language and coding tasks, offering new insights into data requirements for LLM pre-training.

- _Comprehensiveness_: The authors carry out a broad set of experiments, exploring various scales of model size, data quantity, and training stages. Their work provides actionable guidance for future LLM development.

**Weaknesses:**

1. **Limited Details on High-Quality Code Data**: A significant portion of the paper’s conclusions, particularly in Section 3.4, emphasize the positive impact of incorporating high-quality code data on LLM performance across different task categories. However, the description of the high-quality code data is limited to that it is proprietary and synthetically generated. The lack of transparency about the dataset's characteristics, such as the source, data format, or complexity raises questions about what specifically constitutes "high quality" and how these qualities directly influence performance. More details on criteria for data quality would strengthen the paper and enable the research community to apply similar quality standards in future work.

2. **Potential Bias in Generative Ability Evaluation**: The evaluation of generative ability relies on the LLM-as-a-Judge approach, specifically through win-rate scores on the Dolly-200-English dataset, which is designed to reflect open-ended, non-code tasks. The authors attribute improvements in generative quality to the inclusion of code data in pre-training, arguing that code data enriches the model’s ability to generate preferred outputs. However, using LLMs to evaluate other LLM outputs has known limitations[1], including potential biases toward verbosity, structured formatting, and positional advantage. Moreover, the inclusion of markup-style code may enhance readability and formatting, making responses more appealing to the LLM evaluator. To mitigate ambiguity, qualitative examples of output differences would be helpful in illustrating how the inclusion of code data contributes to generative improvements beyond format and readability alone.

3. **Minor Issues**:
   - In Section 3.2, Figure 3 is referenced incorrectly as Figure 8, which could confuse readers.
   - The legend for Figure 8 lacks a label for the gray bar, which reduces clarity in interpreting the data presented.

[1] Zheng, L., Chiang, W. L., Sheng, Y., Zhuang, S., Wu, Z., Zhuang, Y., ... & Stoica, I. (2023). Judging llm-as-a-judge with mt-bench and chatbot arena. Advances in Neural Information Processing Systems, 36, 46595-46623.

**Questions:**

- Could the authors clarify whether the Balanced LM (50% text and 50% code) incorporates markup-style data in its training mix? The paper specifies that the Code-initialized model includes 80% code data and 20% markup-style data, but it remains unclear whether this type of data is also included in the Balanced LM.

---

> ### Author Response · Authors · 2024-11-20
> **Rebuttal by authors**
>
> We would like to thank **R YwMY** for their very positive review of our work, highlighting the novelty and comprehensiveness of the study. We greatly appreciate their claim that our work is _“first-of-its-kind, comprehensive study on how code data in pre-training affects LLMs”_ and noting that our work _“provides actionable guidance for future LLM development”_.  We address each individual concern below:
>
>
> **> Limited Details on High-Quality Code Data: A significant portion of the paper’s conclusions, particularly in Section 3.4, emphasize the positive impact of incorporating high-quality code data on LLM performance across different task categories. However, the description of the high-quality code data is limited to that it is proprietary and synthetically generated. The lack of transparency about the dataset's characteristics, such as the source, data format, or complexity raises questions about what specifically constitutes "high quality" and how these qualities directly influence performance. More details on criteria for data quality would strengthen the paper and enable the research community to apply similar quality standards in future work.**
>
> We thank **R YwMY** for suggesting furnishing more details about what specifically constitutes “high quality” code data. Our primary focus in the given work is to systematically study the impact of code data on downstream tasks across (1) the impact of code across different stages of pre-training, (2) the specific properties of code that contribute to improvement, and (3) how these effects scale. Our synthetic code data was created using problem statements which were used to create Python solutions that are formally verified. The main criteria to create good synthetic code data is to use a performant teacher model to generate the code data and  couple with formal verification tests to ensure generation is high quality and correct. This is similar to recent work that has shown the importance of teachers [1] and formal verification [2].
>
> [1] Odumakinde, Ayomide, et al. "Multilingual arbitrage: Optimizing data pools to accelerate multilingual progress." arXiv preprint arXiv:2408.14960 (2024).
> [2] Xin, Huajian et al. “DeepSeek-Prover-V1.5: Harnessing Proof Assistant Feedback for Reinforcement Learning and Monte-Carlo Tree Search.” ArXiv abs/2408.08152 (2024).
>
> **> Potential Bias in Generative Ability Evaluation: The evaluation of generative ability relies on the LLM-as-a-Judge approach, specifically through win-rate scores on the Dolly-200-English dataset, which is designed to reflect open-ended, non-code tasks. The authors attribute improvements in generative quality to the inclusion of code data in pre-training, arguing that code data enriches the model’s ability to generate preferred outputs. However, using LLMs to evaluate other LLM outputs has known limitations[1], including potential biases toward verbosity, structured formatting, and positional advantage. Moreover, the inclusion of markup-style code may enhance readability and formatting, making responses more appealing to the LLM evaluator. To mitigate ambiguity, qualitative examples of output differences would be helpful in illustrating how the inclusion of code data contributes to generative improvements beyond format and readability alone.**
>
> We thank **R YwMY** for their thoughtful comments about the potential biases in LLM-as-a-judge evaluations. We acknowledge the well-documented limitations of using LLMs to evaluate other LLM outputs, including potential biases toward verbosity, formatting, and positional advantages. In fact, these known limitations were precisely why we included LLM-as-a-judge evaluation as one of multiple evaluation axes in our study. By incorporating this method alongside other rigorous benchmarks, we aimed to provide a holistic and nuanced picture of model performance. We take **R YwMY**’s suggestion to include qualitative examples of output differences in illustrating how there are improvements beyond format and readability.
>
> **> Could the authors clarify whether the Balanced LM (50% text and 50% code) incorporates markup-style data in its training mix? The paper specifies that the Code-initialized model includes 80% code data and 20% markup-style data, but it remains unclear whether this type of data is also included in the Balanced LM.**
>
> We thank **R YwMY** for their question. The Balanced LM’s code data includes markup style data in the same proportion as the code-initialized model, i.e. the 50% code data is made up of 80% code and 20% markup style data (therefore, 40% code and 10% markup data in full mixture). We will add this clarification to the manuscript.
>
> **> Minor Issues**
>
> We thank **R YwMY**  for these suggestions, and acknowledge these issues. These will be updated in the final manuscript.

---

### Official Review · Reviewer_ujyE · 2024-11-02

**Soundness:** 1
**Presentation:** 2
**Contribution:** 1
**Rating:** 3
**Confidence:** 4

**Summary:**

This paper investigates the impact of code data in pre-training for NLP tasks. The authors pre-train Transformer models with sizes ranging from 470M to 2.8B parameters with ablations of code and non-code data. The pre-trained models are evaluated on downstream tasks such as NL reasoning, world knowledge, and code benchmarks. Experimental results show that incorporating code into pre-training has a positive impact on non-coding tasks. In addition, improvements to code quality have an outsized impact across all tasks. In particular, compared to text-only pre-training, the addition of code results in up to relative increase of 8.2% in natural language (NL) reasoning, 4.2% in world knowledge, 6.6% improvement in generative win-rates, and a 12x boost in code performance respectively. The results suggest future investments in code quality and preserving code during pre-training.

**Strengths:**

• The paper provides insights into the effects of code data on pre-training across a range of NLP tasks.

• Experiments are conducted with multiple configurations, such as balancing code and text data, adding depth to the analysis.

**Weaknesses:**

• The findings are somewhat unsurprising, as incorporating diverse data sources (like code) in pre-training is known to improve model performance—a principle demonstrated by previous studies, including PaLM (Chowdhery et al., 2022), Gopher (Rae et al., 2022), and Bloom (Workshop et al., 2023). This paper extends these findings by analyzing code quality and proportion but does not significantly depart from established conclusions, such as the importance of data quality in pre-training (https://arxiv.org/abs/2207.05579).

• Figure 4 suggests that including code data may actually reduce performance on some natural language tasks, which appears contradictory to the paper’s main claim. This inconsistency weakens the overall reliability of the conclusions.

• The pre-training models used in this study are relatively small (470M and 2.8B parameters), which limits the generalizability of the findings. Larger models may be necessary to substantiate conclusions regarding the effects of code data on NLP pre-training.

**Questions:**

1. Given the observed performance drop on certain NL tasks, how does the study reconcile this finding with the broader claim of code data’s benefits for NLP tasks?

2. Have the authors considered evaluating the pre-trained models on additional natural language tasks to better understand the generalization effects of code data?

---

> ### Author Response · Authors · 2024-11-20
> **Rebuttal by authors - Part 1**
>
> We would like to thank **R ujyE** for the time and effort for their feedback and for highlighting the insights provided and the multiple experimental configurations of our work. We address each individual concern below:
>
> **> The findings are somewhat unsurprising, as incorporating diverse data sources (like code) in pre-training is known to improve model performance—a principle demonstrated by previous studies, including PaLM (Chowdhery et al., 2022), Gopher (Rae et al., 2022), and Bloom (Workshop et al., 2023). This paper extends these findings by analyzing code quality and proportion but does not significantly depart from established conclusions, such as the importance of data quality in pre-training (https://arxiv.org/abs/2207.05579).**
>
> While previous works [1][2][3] have noted correlations between code pre-training and improved performance, none have conducted systematic ablations to understand the mechanisms and extent of this relationship. For example, in PaLM, they only noted the use of code data from Github together with its percentage in their data mixture without any systematic analysis and ablation. This is similar to Gopher and Bloom papers. Our work is a first-of-its-kind which makes distinct contributions by systematically investigating how and when code data improves model performance.
>
> Our study is the first to comprehensively examine (1) the impact of code across different stages of pre-training such as pre-training from scratch, continued pre-training and cooldown, (2) the specific properties of code that contribute to improvement such as use of code adjacent data, or high-quality synthetic data, and (3) how these effects scale. Rather than simply confirming that code helps, we provide detailed insights into when and how it helps, including comprehensively quantifying its impact on specific capabilities like natural language reasoning and world knowledge. This systematic study across multiple different model development axes and provides insights for model development at the pre-training stage.
>
> [1] Chowdhery, Aakanksha, et al. "Palm: Scaling language modeling with pathways." Journal of Machine Learning Research 24.240 (2023): 1-113.
> [2] Rae, Jack W., et al. "Scaling language models: Methods, analysis & insights from training gopher." arXiv preprint arXiv:2112.11446 (2021).
> [3] Muennighoff, Niklas, et al. "Scaling data-constrained language models." Advances in Neural Information Processing Systems 36 (2023): 50358-50376.
>
>
> **> Figure 4 suggests that including code data may actually reduce performance on some natural language tasks, which appears contradictory to the paper’s main claim. This inconsistency weakens the overall reliability of the conclusions; [and] Question 1: Given the observed performance drop on certain NL tasks, how does the study reconcile this finding with the broader claim of code data’s benefits for NLP tasks?**
>
> We would like to clarify to **R ujyE** that Figure 4 actually shows the impact of varying percentage of code data in pre-training: as the proportion of code is increased in pre-training while there is indeed a reduction in performance on world knowledge tasks, we consistently observe improvements in natural language reasoning tasks till about >50% of code proportions. This distinction is important and helps illuminate the different ways code data influences model capabilities.
>
> The apparent reduction in world knowledge tasks performance which contain benchmarks TriviaQA and Natural Questions Open. These benchmarks primarily test knowledge retrieval rather than reasoning capabilities. This is not surprising, as code data would not be expected to enhance a model's ability to recall specific facts about world knowledge. As the proportion of code is increased there are no data sources to acquire the required knowledge in the pre-training mix, resulting in degradation of performance. However, considering the trade-off including reasoning tasks, code generation tasks and world knowledge, using the right amount of the code data leads to better average performance overall.
>
> We would also like to point out that when continual-pretraining from a model that has seen code data (Section 3.1, balanced -> text model), we find that there is a relative improvement of 4.1% over a text-only base model even for the world knowledge task.
> Therefore, use of code data with the right amount and combined with multiple stage of pre-training strategy benefits for NLP tasks.

---

> > ### Comment · Reviewer_ujyE · 2024-11-22
> > **Reply to the Rebuttal Part1**
> >
> > - As the authors acknowledge, there have been similar findings in previous works.  The authors argue that this is the first systematic ablation study to understand the mechanisms. However, I do not find the study to be systematic by ablating a few high-level experimental settings. Second, given the current studies, readers cannot understand the mechanism behind the correlations. The paper lacks detailed theoretical discussions on why and how specific properties of code data influence performance on different tasks. Overall, the novelty and contribution of this paper are insufficient to qualify ICLR.
> >
> > - The reduction of performance indicates that the findings are inconsistent across different benchmarks. While the authors attribute this to the task-specific trade-off, the current paper does not adequately analyze when and how their findings can be applied to specific tasks. This lack of clarity weakens the significance of this study.

---

> > > ### Author Response · Authors · 2024-11-25
> > > **Reply to reply for Rebuttal Part1**
> > >
> > > Thank you for your comments. We would like to address each concern below and provide important clarifications.
> > >
> > > > As the authors acknowledge, there have been similar findings in previous works. The authors argue that this is the first systematic ablation study to understand the mechanisms. However, I do not find the study to be systematic by ablating a few high-level experimental settings. Second, given the current studies, readers cannot understand the mechanism behind the correlations. The paper lacks detailed theoretical discussions on why and how specific properties of code data influence performance on different tasks. Overall, the novelty and contribution of this paper are insufficient to qualify ICLR.
> > >
> > > We would like to clarify that, as mentioned in our first response, any of the work R ujyE mentioned, do not provide any findings regarding the impact of code data used in pre-training. They do not explore using the different proportions of code data, or ablate any properties of code used.
> > >
> > > Significantly different from those works, we establish systematic ablations to exhaustively evaluate the impact of code. We evaluate all the below ablations on a diverse set of tasks that includes: NL reasoning, world knowledge, code evaluation, and open-ended generation.
> > >  - **Model initialization**: We explore different initializations of pre-trained models and show that initialization with a code pre-trained model results in improved performance. (Section 3.1 and Figure 2)
> > > - **Model scale**: We scale the model from 470M to 2.8B and show that the same trends hold (Section 3.2, and Figure 3).
> > > - **Proportion of code in pre-training**: We train multiple models from scratch with increasing proportions of code data (from 0% to 100%), and show that increasing code proportions has a positive impact on NL reasoning, and Code performance, and a negative impact on World knowledge tasks (Section 3.3 and Figure 4)
> > > - **Code quality and properties**: we show that quality of code (via use of synthetic code data which is verified), along with properties such as markup-style code data, web-based code data and code-adjacent data have varying impact. We provide precise effects, and details for each ablation. (Section 3.4, and Figure 5)
> > > - **Different stages of pre-training**: we show that code has beneficial effects in cooldown across all tasks. (Section 3.5, and Figure 6)
> > >
> > > We hope that the above further clarifies to R ujyE our systematic ablations, and the level of detail for each experimental setting.
> > >
> > > > The reduction of performance indicates that the findings are inconsistent across different benchmarks. While the authors attribute this to the task-specific trade-off, the current paper does not adequately analyze when and how their findings can be applied to specific tasks. This lack of clarity weakens the significance of this study.
> > >
> > > We would like to clarify that the results on different tasks do not show an inconsistency in our findings, but rather show the intriguing property of the use of code data in LLM pre-training. As mentioned, we carefully picked evaluation benchmarks and deliberately included world knowledge tasks. The experiment R ujyE mentioned (Figure 4), shows the results from differing proportions of code data in pre-training from scratch. The reduction in performance on world knowledge tasks with increase in code proportion is not surprising, as code data would not be expected to enhance a model's ability to recall specific facts about world knowledge. As the proportion of code is increased there are no data sources to acquire the required knowledge in the pre-training mix, resulting in degradation of performance. However, when we continually pre-train a model initialized from a code-pretrained LM, we find that the continual pre-trained model (code -> text, balanced -> text) maintains high performance across all tasks including world knowledge tasks and outperforms text-only models. Therefore, code data when combined with the right amount of text-only data and used carefully in different stages of pre-training does not lead to a permanent drop even in world-knowledge tasks.

---

> ### Author Response · Authors · 2024-11-20
> **Rebuttal by authors - Part 2**
>
> **> The pre-training models used in this study are relatively small (470M and 2.8B parameters), which limits the generalizability of the findings. Larger models may be necessary to substantiate conclusions regarding the effects of code data on NLP pre-training.**
>
> Thanks for your comments. However, our choice of 470M and 2.8B parameter models was deliberate given the comprehensive nature of our study. To conduct thorough ablations and ensure reliable conclusions, we pre-trained 64 models in total where each pre-training run for 200B tokens takes 4,736 TPU-chip hours for 470M and 13,824 TPU-chip-hours for 2.8B parameter models and each cooldown run for 40B tokens takes 1,024 TPU-chip hours for 470M models - a significant computational undertaking even at these scales. Pre-training this many variants at larger scales would be prohibitively expensive.
>
> While we agree that exploring larger architectures would provide further confirmation, as we experimented with two model scales (470M and 2.8B) and found the same trends hold for both scales, we believe that our findings transfer to larger models.  Our findings offer valuable and actionable insights about the role of code in pre-training data mixtures.
>
> **> Have the authors considered evaluating the pre-trained models on additional natural language tasks to better understand the generalization effects of code data?**
>
> We thank **R ujyE** for their suggestion. We evaluated our models on 14 natural language datasets where these datasets cover a diverse set of task types for reasoning and world knowledge such as question answering, natural language inference, sentence completion, and coreference resolution. Additionally, we included an open-ended generation task to validate our conclusions beyond academic benchmarks. Finally, we included 2 code datasets as given in Table 1 (Appendix). This is a very comprehensive list in terms of tasks types, datasets, domain to enable consistent evaluation signal at our chosen model scales, allowing us to draw reliable conclusions about the effects of code data.
>
> Additionally, thanks to **R PKei** suggestions for exploring model performance on mathematical tasks, we take the additional time during this rebuttal period to evaluate the cooldown 470M models on GSM8K. The results show that including code in any stage of the pre-training improves performance compared to the model where no code has been seen pre-training, continual pre-training, and cooldown. The most performant model in this comparison has seen the code data in all stages including cooldown where it leads a significant improvement (from 2.9 to 4.12, +42% relative gain).
>
> Finally, it is important to note that, we carefully picked these benchmarks by filtering out additional benchmarks where any experimented model performance was below random. Therefore, we made sure that all the experimental models’ performance achieves non-trivial performance in selected benchmarks.

---

> ### Author Response · Authors · 2024-11-22
> **Request for response**
>
> As the discussion period has only a few days remaining, we wanted to ask **R ujyE** if there are any follow-up points we can clarify. We have responded to all concerns raised, and also included new results on  a mathematical reasoning benchmark (GSM8K) where we show that code data plays a beneficial role as it is used in different stages of pre-training. We are also happy to engage in further discussion. If there are no other points of clarification, we would kindly ask that Reviewer **ujyE** to consider increasing their score to reflect.

---

> ### Comment · Reviewer_ujyE · 2024-11-22
> **Reply to the Rebuttal Part2**
>
> Thank you for your rebuttal. While your response has explained the two questions I raised, I remain unconvinced based on the current answers:
>
> - Given the computational resources available for this study (e.g., 13,824 TPU-chips), I think experimenting with larger models (e.g., 7B) is not prohibited. Even conducting a subset of the main experiments on larger models would provide more robust insights.
>
> - The authors argue that the paper has been evaluated on 14 datasets, but why are the most conventional text classification benchmarks excluded, which are naturally the first choice to evaluate LLM capabilities?

---

> > ### Author Response · Authors · 2024-11-25
> > **Reply to reply for Rebuttal Part2**
> >
> > Thank you for your comments. We would like to address each concern below and provide important clarifications.
> >
> > > Given the computational resources available for this study (e.g., 13,824 TPU-chips), I think experimenting with larger models (e.g., 7B) is not prohibited. Even conducting a subset of the main experiments on larger models would provide more robust insights.
> >
> > We would like to clarify to R ujyE that 13,824 is not TPU-chips but TPU-chip-**hours** for each 200B tokens pre-training at 2.8B scale which roughly corresponds to **52 hours wall-clock time**. Further, each experiment at 2.8B scale itself involves a significant cost given the nature of pre-training (to be transparent, it corresponds to roughly $16K), but a 7B run with the same amount of tokens (which would not be optimal due to higher data requirement in that scale) requires about 520 hours wall-clock time and 10x cost compared to 2.8B experiments.
> >
> > **Therefore scaling experiments to a 7B model would go much beyond our research budget.** Furthermore, considering that for this research we pre-trained 64 models, the cost of one 7B run which corresponds to approximately ten 2.8B runs and twenty 470M runs, would **significantly** narrow the scope of our systematic study.
> >
> > We would like to emphasize that our findings are consistent across both 470M and 2.8B scales, therefore, we believe the same trends would hold at larger-scale LLMs.
> >
> > > The authors argue that the paper has been evaluated on 14 datasets, but why are the most conventional text classification benchmarks excluded, which are naturally the first choice to evaluate LLM capabilities?
> >
> > We carefully selected our evaluation datasets based on reviewing evaluation benchmarks used in multiple works [1], [2], [3] to evaluate similar scale LLMs, which are all published in major ML conferences. Therefore academic benchmarks used in our research and these papers are heavily shared. We believe this is a good indication that our evaluation setting is reliable. We additionally added code benchmarks and evaluation of generative performance using LLM-as-a-judge to have a more comprehensive view of LLMs’ performance experimented in our paper.
> >
> >
> > [1] Muennighoff, Niklas, et al. "Scaling data-constrained language models." Advances in Neural Information Processing Systems 36, Best Paper (Neurips 2023).
> > [2] Stella Biderman, et al. “Pythia: a suite for analyzing large language models across training and scaling.” In Proceedings of the 40th International Conference on Machine Learning (ICML 2023).
> > [3] Ahmadian, Arash, et al. "Intriguing properties of quantization at scale." Advances in Neural Information Processing Systems 36 (Neurips 2023)

---

### Official Review · Reviewer_SkF7 · 2024-11-04

**Soundness:** 3
**Presentation:** 3
**Contribution:** 3
**Rating:** 6
**Confidence:** 3

**Summary:**

The authors examine the role of code data in the pre-training of large language models (LLMs), a practice that has become increasingly common, even for models not specifically designed for code. Despite anecdotal consensus among practitioners regarding the importance of code in enhancing the performance of general LLMs, there has been limited research analyzing its precise impact on non-code tasks. This study systematically investigates the influence of code data in pre-training on a diverse array of downstream tasks beyond code generation. The authors pose the question: “What is the impact of code data used in pre-training on a large variety of downstream tasks?” Through extensive ablation studies, they evaluate models ranging from 470M to 2.8B parameters across natural language reasoning tasks, world knowledge tasks, code benchmarks, and LLM-as-a-judge win rates. The results consistently demonstrate that code serves as a critical building block for generalization well beyond coding tasks. Improvements in code quality significantly enhance overall task performance, with findings indicating up to an 8.2% relative increase in natural language reasoning, a 4.2% boost in world knowledge, a 6.6% improvement in generative win rates, and a 12x enhancement in code performance when compared to text-only pre-training. The authors suggest that investing in code quality and retaining code during pre-training can yield substantial benefits across various tasks.

**Strengths:**

+ Important Area.
+ Interesting Finding.
+ Analysis of the Code Quality is Good.

**Weaknesses:**

- Technical Contribution is Weak.
- Limited Scale.

**Questions:**

The investigation into the role of code in LLM pre-training addresses a crucial aspect of model performance with good implications for both research and practical applications. The exploration of the impact of code data and strategies for leveraging it provides valuable insights into enhancing model effectiveness, contributing to a deeper understanding of LLM capabilities.

I have two concerns:

1. Limited Technical Contribution: The study primarily consists of empirical analyses without introducing new methodologies or frameworks. While the findings are good, the technical contribution could be perceived as limited due to the absence of novel approaches or theoretical advancements.

2. Limited Scale: The analysis is restricted to models with 470M and 2.8B parameters, which may not capture the performance dynamics across a broader range of model sizes and architectures.

---

> ### Author Response · Authors · 2024-11-20
> **Rebuttal by authors**
>
> We would like to thank **R SkF7** for the time and effort for their positive feedback and for highlighting the importance of the area studied, and the analysis of code quality. We also express gratitude towards them for emphasizing positively that our approach, _“addresses a crucial aspect of model performance with implications for both research and practice”_ and that our approach _“provides valuable insights into enhancing model effectiveness”_. We address the individual concerns below:
>
> **> Limited Technical Contribution: The study primarily consists of empirical analyses without introducing new methodologies or frameworks. While the findings are good, the technical contribution could be perceived as limited due to the absence of novel approaches or theoretical advancements.**
>
> Thanks for your comments. While we do not introduce new methodological frameworks, we want to respectfully clarify that our research fills a critical knowledge gap through systematic empirical investigation of the impact of code in LLM pre-training. Despite the widespread practice of including code in pre-training data mixtures for LLMs, there has been limited rigorous analysis of its impact on non-code tasks before our work. While theoretical advancements are valuable, equally important are rigorous empirical studies that challenge assumptions and provide concrete evidence for improving model and data design. Our work falls into this latter category, offering actionable insights about the critical role of code data during pre-training and its effects on downstream non-code tasks. Considering the wide-spread use of LLMs and the cost of pre-training, we believe that our work results in significant and impactful research.
>
> **> Limited Scale: The analysis is restricted to models with 470M and 2.8B parameters, which may not capture the performance dynamics across a broader range of model sizes and architectures.**
>
> We deliberately chose the model scales of 470M and 2.8B parameters, given the computational demands of our experimental design. We pre-train 64 models in total where each pre-training run for 200B tokens takes 4,736 TPU-chip hours for 470M and 13,824 TPU-chip-hours for 2.8B parameter models and each cooldown run for 40B tokens takes 1,024 TPU-chip hours for 470M models. This is an enormous endeavour given the scale and computational resources required.
>
> While we agree that exploring larger architectures would provide further confirmation, as we experimented with two model scales (470M and 2.8B) and found the trends hold for both scales, we believe that our findings transfer to larger models.

---

> > ### Comment · Reviewer_SkF7 · 2024-11-21
> > **Thanks**
> >
> > Thanks for the responses. They're good, but the issues remain, so I'll maintain my current rating.

---

> ### Author Response · Authors · 2024-11-22
> **Acknowledgment of response**
>
> We thank **R SkF7** for acknowledging the rebuttal and their comments.

---

### Official Review · Reviewer_PKei · 2024-11-04

**Soundness:** 3
**Presentation:** 3
**Contribution:** 3
**Rating:** 6
**Confidence:** 4

**Summary:**

The paper investigates the impact of incorporating code data into pre-training for large language models (LLMs).
It explores how code data influences various tasks, including natural language reasoning, world knowledge, and open-ended generation.
Through extensive ablation studies, the authors reveal that adding code data yields substantial benefits across tasks beyond coding.
The study emphasizes code quality and finds that synthetic, high-quality code data further enhances general performance.
The results imply that code data should be included in the pre-training process as it significantly enhances generalization across tasks.

**Strengths:**

- The paper addresses a crucial question: does code data improve performance on non-code tasks? This question has been relatively unexplored, and the paper offers a fresh perspective on the potential benefits of code data for models primarily intended for natural language tasks.

- The methodology is rigorous, employing controlled experiments that examine various factors, such as code quality, proportions, and stages of pre-training. The large-scale experiments (up to 2.8B parameters) provide an in-depth understanding of how code data impacts model performance.

- This work has significant implications for pre-training dataset design, suggesting that code data improves generalization and that high-quality synthetic code can be particularly impactful. This insight can guide future LLM development, especially in choosing and refining pre-training data.

**Weaknesses:**

- For the scaling experiments, some key information could be explored further. For instance, does the trend observed in Figure 4 hold consistently across models of different scales? Additionally, how can the trends identified be extended or generalized to larger models?

- I am also curious about the model's performance on mathematical tasks, as these, like coding, are highly logical and reasoning-intensive. Including a benchmark evaluation on mathematical tasks could provide valuable insight into whether similar gains extend to other logic-focused domains.

- In terms of writing, certain critical details could be clarified. For instance, it may not be immediately clear in the text which model size (e.g., 470M or 2.8B) is being evaluated in figures such as Figures 4 and 5. Adding such context in the captions or main text would enhance clarity for readers.

**Given the close relationship between scaling laws and research on dataset composition, there is still room to refine and expand upon some of the paper's exploratory findings.**

**Questions:**

- In the sentence "Figure 8 shows the results of xxx" in Section 3.2, it seems that "Figure 8" should actually be "Figure 3." May be a typo?

---

> ### Author Response · Authors · 2024-11-20
> **Rebuttal by authors**
>
> We would like to thank **R PKei** for their positive feedback and for highlighting the _“extensive ablation study”_ that shows that _“adding code data yields substantial benefits across tasks beyond coding”_. We also express gratitude towards them for emphasizing positively that our approach, _“addresses a crucial, relatively unexplored question”_, _“offers a fresh perspective on potential benefits of code data”_ and that _“the methodology is rigorous”_, _“provides significant implications for pre-training dataset design”_. We address the individual concerns below:
>
> **> For the scaling experiments, some key information could be explored further. For instance, does the trend observed in Figure 4 hold consistently across models of different scales? Additionally, how can the trends identified be extended or generalized to larger models?**
>
> Thanks for your comment. For Figure 4, we pre-trained 6 models by varying code percentage before continual pre-training. Given the resource intensive nature of pre-training (requires training 6 larger models for this experiment), we have only conducted this ablation using 470M parameters models (pre-trained for 200B tokens).
>
> However, we believe that a similar trend would hold for the larger models. In the context of continued pre-training experiments (Section 3.2), we pre-train 2.8B parameters models where we see a similar trend with 470M models where the code initialized model (i.e. code -> text) outperforms the balanced initialized model (balanced -> text) in reasoning tasks, and falls slightly behind in world knowledge tasks.
>
>
> **> I am also curious about the model's performance on mathematical tasks, as these, like coding, are highly logical and reasoning-intensive. Including a benchmark evaluation on mathematical tasks could provide valuable insight into whether similar gains extend to other logic-focused domains.**
>
> **R PKei** suggestions around exploring model performance on mathematical or logic-focused tasks are insightful. We take the additional time during this rebuttal period to evaluate the cooldown 470M models on GSM8K. We only evaluate the models that underwent cooldown training as for this model size, only after cooldown, the models achieve nontrivial performance on GSM8K.
> | | Text-Only (no-code cooldown) | Text-Only (code cooldown) | Balanced -> text (no-code cooldown) | Balanced -> text (code cooldown) |
> |---|---|---|---|---|
> | GSM8K (5-shot, maj@1) | 2.9 | 3.6 (+24%) | 4.05 (+40%) | 4.12 (+42%) |
>
> As can be seen from the results, including code in any stage of the pre-training improves performance compared to the model where no code has been seen pre-training, continual pre-training and cooldown. The most performant model in this comparison has seen the code data in all stages including cooldown where it leads a significant improvement (from 2.9 to 4.12 - +42% relative gain).
>
>
> **> In terms of writing, certain critical details could be clarified. For instance, it may not be immediately clear in the text which model size (e.g., 470M or 2.8B) is being evaluated in figures such as Figures 4 and 5. Adding such context in the captions or main text would enhance clarity for readers.**
>
> We thank **R PKei** for pointing out improvement required in clarity for certain details. We will update the main text, captions and related text by adding the information of the model size (470M parameters) for Figures 4 and 5 to improve readability and provide clear details.
>
> **> In the sentence "Figure 8 shows the results of xxx" in Section 3.2, it seems that "Figure 8" should actually be "Figure 3." May be a typo?**
>
> Thanks for finding this typo. Yes, it should be Figure 3, we have corrected this in Section 3.2.

---

> ### Author Response · Authors · 2024-11-22
> **Request for response**
>
> As the discussion period has only a few days remaining, we wanted to ask **R PKei** if there are any follow-up points we can clarify. As suggested by **PKei**, we have evaluated our cooldown experimental variants on a mathematical reasoning benchmark (GSM8K), and show that code data plays a beneficial role as it is used in different stages of pre-training. We have responded to all concerns raised, and are also happy to engage in further discussion. If there are no other points of clarification, we would kindly ask that Reviewer **PKei** consider increasing their score to reflect.

---

> > ### Author Response · Authors · 2024-12-02
> > **Final request for response**
> >
> > We would like to again thank R PKei for their constructive feedback which is now incorporated in the final manuscript. As the discussion period has only a day remaining, we wanted to ask R PKei if there are any follow-up points we can clarify. If there are no other points of clarification, we would kindly ask that Reviewer PKei consider increasing their score to reflect.

---

### Author Response · Authors · 2024-11-20
**General response to reviewers**

We greatly appreciate the thoughtful and positive feedback from the reviewers. To the best of our knowledge, our work is first to systematically study the impact of code data in LLM pretraining, with critical ablations focusing on (1) stages of pretraining (2) properties of code, and (3) scale. As described by **R YwMY** our work is “a first-of-its-kind, comprehensive study on how code data in pre-training affects LLMs across various natural language and coding tasks, offering new insights into data requirements for LLM pre-training”.

We are encouraged that reviewers found the question is critical and “relatively unexplored, and the paper offers a fresh perspective on the potential benefits of code data” [**PKei**],  “Important Area” [**SkF7**].  **PKei** commended our "rigorous methodology," highlighting our controlled experiments examining code quality, proportions, and pre-training stages. Our large-scale experiments up to 2.8B parameters were recognized as providing an in-depth understanding of code data's impact on model performance [**Pkei**, **ujyE**]. The reviewers saw significant potential in our findings for guiding future LLM development, particularly in pre-training dataset design. Reviewers specifically noted our work has "significant implications" [**PKei**] for understanding how code data can improve generalization.

We also thank the reviewers for their constructive feedback. We have provided detailed explanations addressing each of their points including additional evaluation, demonstrating how our work contributes meaningfully to the understanding of the role of the code data in language model pre-training. We look forward to engaging in meaningful discussion and welcome any additional questions at any point in time.

---

### Author Response · Authors · 2024-12-02
**Final General Response to Reviewers**

After carefully reading the comments and suggestions from all reviewers, we have made the following updates over the course of the rebuttal period, in addition to meticulously responding to all the reviewer’s concerns:
- Thanks to R PKei's suggestion, we evaluate the 470M parameters models which include cooldown training phase, on GSM8K. We find that having code data in any stage of the pre-training improves performance compared to the model where no code has been seen throughout pre-training, continual pre-training or cooldown. The most performant model in this comparison has seen the code data in all stages including cooldown where it leads a significant improvement (from 2.9 to 4.12, +42% relative gain).
- We have improved the readability of the manuscript, correcting errors pointed out by R PKei and R YwMY.

All updated changes are reflected in the final manuscript (we were unable to upload the revision here). Additionally, we carefully addressed the main concerns about the model scale highlighting the exhaustive set of pretraining experiments in two parameter sizes where we see the same trend across 470M and 2.8B models, and the evaluation suite showcasing the comprehensive nature and rigor of our analysis.

---

### Meta-Review · Area_Chair_TirG · 2024-12-22

**Metareview:**

This paper investigates an important question of whether code data can be used to improve performance on non-code tasks and has been well-received by all reviewers. The reviewers unanimously agree that the results are impressive, demonstrating significant contributions to the field. Congratulations to the authors on the acceptance of this paper.

**Additional Comments On Reviewer Discussion:**

This problem has been well-received by all reviewers. The reviewers unanimously agree that the results are impressive, demonstrating significant contributions to the field.

---

### Decision · Program_Chairs · 2025-01-22

Accept (Poster)